# Broadly Neutralizing Bovine Antibodies: Highly Effective New Tools against Evasive Pathogens?

**DOI:** 10.3390/v12040473

**Published:** 2020-04-22

**Authors:** Matthew J. Burke, Peter G. Stockley, Joan Boyes

**Affiliations:** 1School of Molecular and Cellular Biology, Faculty of Biological Sciences, University of Leeds, Leeds LS2 9JT, UK; bs14m4b@leeds.ac.uk (M.J.B.); p.g.stockley@leeds.ac.uk (P.G.S.); 2Astbury Centre for Structural Molecular Biology, Faculty of Biological Sciences, University of Leeds, Leeds LS2 9JT, UK

**Keywords:** broadly neutralizing antibodies, ultralong CDR H3, bovine antibodies, antibody diversity

## Abstract

Potent antibody-mediated neutralization is critical for an organism to combat the vast array of pathogens it will face during its lifetime. Due to the potential genetic diversity of some viruses, such as HIV-1 and influenza, standard neutralizing antibodies are often ineffective or easily evaded as their targets are masked or rapidly mutated. This has thwarted efforts to both prevent and treat HIV-1 infections and means that entirely new formulations are required to vaccinate against influenza each year. However, some rare antibodies isolated from infected individuals confer broad and potent neutralization. A subset of these broadly neutralizing antibodies possesses a long complementarity-determining 3 region of the immunoglobulin heavy chain (CDR H3). This feature generates unique antigen binding site configurations that can engage conserved but otherwise inaccessible epitope targets thus neutralizing many viral variants. Remarkably, ultralong CDR H3s are a common feature of the cow antibody repertoire and are encoded by a single variable, diversity, joining (VDJ) recombination that is extensively diversified prior to antigen exposure. Recently, it was shown that cows rapidly generate a broadly neutralizing response upon exposure to HIV-1 and this is primarily mediated by these novel ultralong antibody types. This review summarises the current knowledge of these unusual CDR H3 structures and discusses their known and potential future uses.

## 1. Introduction

Jawed vertebrates generate millions of B cells every day, each of which expresses a unique antigen receptor, either on the cell surface as a B-cell receptor (BCR) or secreted in the form of an antibody (Ab). These receptors consist of two identical light chains and two identical heavy chains joined by non-covalent interactions and disulphide bonds. Each individual polypeptide chain has an N-terminal variable domain and a C-terminal constant domain, where the constant domain of the heavy chain determines the Ab isotype (IgM, IgD, IgG, IgA or IgE). The variable domain is unique to each BCR and it is this region that contacts the antigen, determining the Ab specificity [1]. The variable domains of the heavy and the light chains each contains four framework regions (FRs) and three complementarity-determining regions (CDRs). The FRs are associated with structural stability and act as a scaffold for the Ab structure, whereas the CDRs are hyper-mutable and form the antigen binding site (paratope) that interacts with a specific region of the antigen (epitope). The CDR 3 of the heavy chain (CDR H3) typically forms a large fraction of the antigen binding pocket [2,3].

Antibody diversity is fundamental to eliminating the array of pathogens that the host could encounter. The assembly of the pre-immune Ab repertoire begins in pro-B cells within the bone marrow when one from each of the variable (V), diversity (D) and joining (J) gene segments is assembled into a contiguous variable exon at the heavy chain locus (IgH) by V(D)J recombination. This process is catalysed by the lymphoid-specific recombination activating gene (RAG) 1 and RAG 2 proteins (Figure 1). RAG-mediated cleavage is targeted to the V, D and J gene segments by recombination signal sequences (RSSs) that consist of a conserved heptamer and nonamer, separated by a relatively non-conserved spacer of 12 ± 1 or 23 ± 1 base pairs. Following cleavage, the broken ends are shepherded towards the canonical non-homologous end-joining repair pathway for processing and ligation [4,5,6]. Primary diversity is focussed in the sequence encoding the CDR H3 loop and sequence diversity here is increased by non-templated nucleotide additions and exonucleolytic trimming [7,8]. The generation of the variable exon at the light chain loci is analogous, except recombination is between V and J gene segments only.

Exposure to antigen in the germinal centres (GC) of the secondary lymphoid tissues induces the marked diversification of the Ab sequences. This tightly regulated process of somatic hypermutation (SHM) is initiated by activation-induced cytosine deaminase (AID) activity and the recruitment of downstream repair enzymes that are uniquely error-prone at the immunoglobulin loci [9]. Assuming it is functional, the newly mutated Ab, with a novel binding site, is then tested for improved antigen binding affinity as the B cell migrates to the light zone of the GC. The ability of B cells to discriminate the affinity of their unique receptor for an antigen efficiently ensures that advantageous mutations are favoured, with a concomitant improvement in antigen binding [10,11]. Owing to their limited number of Ig gene segments, cows (*Bos taurus*) depend upon this AID-induced SHM to expand their limited pre-immune Ab repertoire prior to antigen exposure [12]. Jointly, these Ab diversification mechanisms generate a vast Ab repertoire. Some viruses, including HIV-1, influenza and Ebola, utilise a multitude of evasive tools to circumvent the host immune system. Some of these pathogens have an enormous variety of globally circulating strains, can rapidly diversify their surface antigens, or occlude epitopes that are evolutionary conserved, to limit their immunogenicity. Such strategies severely limit the effectiveness of typical neutralizing Ab responses [13]. However, it was observed that the Abs from some chronically HIV-1 infected patients exhibit broad and potent neutralization of many viral variants [14]. Improved single-cell isolation, enhanced cloning techniques and the advent of Next Generation Sequencing (NGS) have since facilitated the isolation, generation and subsequent characterization of many of these broadly neutralizing Abs (BNAbs) [15,16]. Notably, one of the most striking features of some BNAbs is a long (≥20 amino acids) CDR H3 [17]. Recently, it has been shown that Abs with ultralong CDR H3s make up a significant fraction of cow Abs [18,19] and an effective BNAb response can be rapidly generated in cows in as little as six weeks [20]. Similarly, the heavy chain only ‘nanobodies’ generated by *Camelidae* utilise a long CDR H3 to bind to hidden epitopes that can neutralize a broad cohort of HIV variants [21,22].

This review focuses on such BNAbs, and in particular on their extended CDR H3, a striking feature that has been shown to facilitate the broad neutralization of HIV-1 and other viruses. While an extended CDR H3 does not automatically confer neutralization properties per se, it does provide a scaffold for unconventional paratope structures that can puncture viral defences, such as thick glycan coats, to engage conserved epitope regions and thus neutralize many variants. We will summarise the current knowledge of these unusual CDR H3 structures in humans and cows, whilst exploring differences in the diversification mechanisms between these species, and discuss future strategies for the utilisation of the novel properties of BNAbs.

## 2. The Long CDR H3 Is Generated Inefficiently in Humans

The most studied BNAbs are those from HIV-1 infected individuals. Although the mean CDR H3 length in human Abs is ≈15 amino acids [23], some anti-HIV-1 BNAb lineages have an extended CDR H3 over 35 amino acids that is capable of reaching hidden epitopes typically associated with limited Ab recognition [16]. The sequence encoding the CDR H3s of any Ab can be considered to begin after a conserved cysteine codon (C104) at the 3′ end of FR3 in the V segment and extends to a conserved W/F residue encoded at the 5′ end of the J segment that marks FR4 (Figure 1; IMGT numbering; [24]). As such, it spans the entire D segment, two recombined junctions and is intrinsically variable. Indeed, several lymphocyte-specific mechanisms have been described that can significantly extend the length of the sequence encoding the CDR H3. The principal ways include extensive junctional diversity and insertion/deletion mutations associated with SHM, but evidence has also been found for D-D gene segment fusions and VH replacement events [17]. Though long CDR H3s are prevalent in the human transitional and naïve B cell Ab repertoire, their frequency is reduced as B cells mature in normal adult repertoires [25]. Indeed, Abs from memory B cells and plasmocytes exhibit a shorter mean CDR H3, which likely indicates that long CDR H3s are not selected during affinity maturation [26]. It therefore seems that the generation of long CDR H3 BNAbs from the antigen-naïve repertoire in humans is inefficient due to relatively few immunogens that target the germline-encoded BCRs, which is compounded by the fact that the unmutated precursors of many mature BNAbs bind only weakly to their relative antigen [27]. Indeed, broad and potent neutralization emerges stochastically in only some individuals and only then after years of chronic infection. Invariably, this requires a co-evolutionary process of sequential exposure to antigen variants together with iterative affinity maturation [15,16,28]. It is notable that many individuals who do generate an effective broadly neutralizing response against HIV-1 have Abs with a longer average CDR H3 [25].

When present, the long CDR H3 lifts the paratope away from the main antibody structure and allows CDRs to adopt complex conformations that include long flexible loops with anionic tips [28,29], high-affinity loop insertions [30] and intricately folded knob domains [18]. Most BNAbs have undergone extensive SHM away from their germline configuration. This SHM frequently involves insertion/deletion (indel) mutations of multiple codons [31], which often foster critical stabilizing interactions between the protruding CDR H3 structure and the other CDRs in human BNAbs [16]. Moreover, as well as stabilizing the protruding structure and directly altering the antigen contacts, SHM can generate local flexibility that allows the antigen-binding site to engage with its respective target at a variety of angles, significantly improving neutralization activity in some cases [16]. Although rare in human antibody genes, large nucleotide insertions (>40 nt) introduced by sequence duplication during the error-prone repair of SHM have been shown to contribute to antigen recognition [32]. Whilst the high SHM rate is not conducive to their induction via classical vaccination, it does not prevent the use of previously generated BNAbs therapeutically [33,34].

Significant advances have been made regarding induction of HIV-1 BNAb precursors in transgenic mice models carrying knock-ins of either the respective BNAbs germline V, D and J segments or the near mature BNAb VDJ exon [35,36]. Further innovative efforts have enabled the use of these novel extended CDR H3 Ab types for prophylaxis and therapy, and similar approaches have shown considerable promise in clinical trials with HIV-1 infected individuals [34]. Expanding the search for BNAbs into the Ab repertoires of diverse species will likely provide new opportunities to develop immunotherapeutic agents.

## 3. Ultralong CDR H3 Are Prevalent in the Cow Ab Repertoire

In contrast to humans, the bovine immune system relies much more extensively on mechanisms that occur after the original recombination event to diversify their immunoglobulin genes [37]. Only 12 IGHV, 16 IGHD and 4 IGHJ gene segments appear to be functional in cows [23], which is significantly less than the >100 V, 25 D and 6 J gene segments found in humans. In cows, the initial lack of diversity is compensated by extensive AID-induced SHM [37], which occurs in the gut-associated lymphoid tissues prior to antigen exposure [12]. While diversification of the pre-immune Ab reservoir has been shown to take place to some extent in the Peyer’s patch of mice [38], this is not a major driver of primary Ab diversity as it is in the cow. 

Cow antibodies can be split into two groups based on their CDR H3 lengths: 1–39 amino acids, and 40–71 amino acids [18,39,40,41]. Recently, a detailed assembly of the bovine *IgH* locus was reported (KT723008) [23] that localised all the functional *IGH* genes to chromosome 21 and also confirmed the presence of a single, ultralong D segment (DH8-2) of 148 bp in length, although longer alleles have been reported [12]. In comparison, the longest D segment in the human *IgH* locus is 37 bp (D3–16), while mouse CDR H3s are typically even shorter than human CDR H3s; clearly, the potential for an ultralong CDR H3 does not exist in the germline of these species [42,43]. Nonetheless, the NGS analysis of bovine antibody repertoires [18] suggests that nearly all of the very long bovine Abs (40–71 amino acid CDR H3) are generated by the diversification of just this one long D8-2 gene segment. In fact, these ultralong CDR H3 sequences utilise the same V (VH1-7) and J (J2-4) gene segments in each case, together with the long DH8-2. Structural studies have shown that the ultralong CDR H3s encode a long, solvent exposed β−strand stalk that protrudes from the main Ab structure and supports a disulphide bonded knob domain that is responsible for antigen binding [18]. Although it has also been reported that insertion of up to 100 nucleotides at the VD or DJ junctions can generate very long CDR H3s from shorter D gene segments in cows [23], it is not known if the resulting antigen receptors are functional. Nonetheless, the bovine immune system is seemingly primed to elicit Abs with ultralong CDR H3s.

## 4. Diversification of the Ultralong CDR H3 in Cows

AID activity plays a significant role in the generation of the bovine ultralong CDR H3 [37,44]. This SHM activity generates large internal deletions (≤50 nucleotides) and relatively large insertions (≤20 nucleotides) within the DH8-2 segment, which diversify the pattern of disulphide bridge formation to act as a rapid means of altering the antigen binding domain [37,44]. Though AID commonly introduces point mutations and small insertion/deletion mutations in the Ig sequences of other species, the deletions in bovine D8-2 are larger and more frequent than those observed in humans or mice [37,38,44]. The overlapping AID targeting hotspots in the D8-2 segment may facilitate the production of staggered double strand breaks (DSBs) by generating two single-strand nicks simultaneously (Figure 2). This is analogous to the way in which AID generates DSBs in the Ig switch regions [9,38] and in non-Ig genes, including the *myc* gene leading to the *myc-IgH* translocation [45]. The mechanisms mediating such large deletions and insertions in D8-2 are currently unknown but may involve aspects of microhomology-mediated end-joining during direct DSB repair, or be the result of template slippage events (Figure 2) [37] across the repetitive D8-2 sequence. Nonetheless, when combined with extensive SHM, starting with a longer germline gene segment that is subsequently trimmed internally is seemingly a very efficient method of producing a diverse antibody repertoire from a single gene [37].

Further diversity in the bovine ultralong CDR H3 is contributed by the unusual, adenine-rich insertions at the V-D junction that are likely deposited during the original V-DJ recombination event [18,19]. These insertions are a significant difference from the TdT-mediated junctional diversity that has been observed in human and mouse heavy chain rearrangements and the details of the bovine insertion mechanism(s) remain unknown. However, due to its presence in the region encoding the stem and its propensity for encoding hydrophilic residues [18], this non-germline sequence can likely alter the orientation of the knob domain while maintaining the β−strand structure. This feature may therefore change the local flexibility and geometry without compromising the overall stem stability [46]. Seven recent structures reveal that small changes (<7°) in the angle of the stalk can lead to large changes in the orientation of the putative antigen binding site [46]. Notably, a threonine is nearly always found at the second position of this junctional insertion and is positioned directly opposite alternating tyrosines (YxYxY) encoded in the germline and present in the descending side of the stem in the 3D structure [18,47]. How changes specific to this V-D junctional sequence affect the binding or stability of the structure has not yet been explored but could feasibly be examined via the alanine-scanning mutagenesis of an Ab with a known antigen target of the ultralong CDR H3. 

## 5. Uncovering the Structure of the Cow Ultralong CDR H3

The ultralong CDR H3 structure was first characterised in detail in two cryo-EM structures [18] and has since been further analysed by Stanfield et al. [47], Dong et al. [46] and reviewed by others [19,48]. Briefly, the stalk of bovine long CDR H3 Abs consists of anti-parallel β- strands that protrude away from the main Ab structure, akin to the strands that stabilize the anionic protrusions of some human BNAbs. At the tip of this stalk is a disulphide-bonded knob domain that is likely the only region that contacts the epitope. The removal of the knob domain completely abrogates Ab binding [18], while its transplantation onto an antibody-scaffold is sufficient to transfer the donor Abs binding specificity with only a moderate loss of neutralization potency [20].

The stabilisation of the ultralong cow CDR H3 does not require SHM, but instead the structure is stabilized by residues encoded by the germline sequence of both the IgH and IgL chains [18,46,47]. The 3′ end of VH1-7 contains a sequence duplication that encodes CTTVHQ, a relatively conserved motif that stabilizes the base of the stalk by interacting with a DKAVG motif of CDR H1 [18,37,47]. All other CDRs, including those of the light chain, act to stabilise the CDR H3 domain but do not participate directly in antigen binding. In fact, amino acid variability in CDR H1 and CDR H2 is remarkably low [37]. The knob region itself lacks a hydrophobic core and is instead compacted and shaped primarily by intra-domain disulphide bonding [18,19,47]. This feature explains the strong preference for an even number of cysteine residues in mature sequences encoding ultralong CDR H3s [18]. Nonetheless, of the four germline cysteines in the ultralong CDR H3, only the first is apparently conserved and denotes the start of the CPDG motif at the start of the D segment [19]. Notably, there is a high density of consensus AID-targeting motifs within the germline D8-2 and the majority of codons are only a single mutation away from encoding a cysteine [18,44]. NGS data have further confirmed that somatic diversity is almost entirely focussed within the CDR H3 [18,44,47], and massively alters the cysteine content and its distribution within the knob domain. This mediates changes to the disulphide bonding pattern, the loop length and the overall architecture of the ultralong CDR H3 to generate a diverse Ab repertoire from each VH1-7, DH8-2 and JH2-4 rearrangement. The 3′ of the D8-2 segment contains a fairly conserved pattern of alternating tyrosines that likely help to support the knob domain [18].

Due to the relative infancy of the research, it is probable that diversity in the bovine ultralong CDR H3 is even more expansive than that which has been characterised to date. Consistent with this, the first five cryo-EM structures containing bovine ultralong CDR H3 regions suggested that the disulphide bonds are restricted to cysteine residues within the globular mini-domain [18,47]. However, in seven more recent structures, Dong et al. [46] identified a new disulphide bonding configuration in which two cysteines in the stem form disulphides with two others in the knob domain. This linkage has the effect of changing the angle of the antigen-binding site to generate further diversity [46]. Consistent with this, the knob domain of the ultralong CDR H3 has an inherent flexibility, as demonstrated by the difficulties in obtaining structures for many of the ultralong loops [46].

There is, however, remarkable conservation elsewhere in the Ab heterodimer. The sequences of the light chain in bovine Abs is a relatively invariable version of IgλV30, so much so that this aspect of the structure is generally indistinguishable from one Ab to another (Figure 3) [18,19,47]. In fact, the Abs used in the biochemical and structural analysis by Wang et al. [18] were all purified with an identical light chain, and two-thirds of CDR H3 sequences in a more recent study were successfully expressed as a FAb with this same light chain [46]. This strongly suggests that light chain pairing is not as strict as in human BNAbs, which will facilitate the creation of ultralong bovine CDR H3 libraries. 

## 6. Long CDR H3 Structures Confer Broadly Neutralizing Anti-Viral Properties

A plethora of BNAbs have now been isolated and cloned from human HIV-1 patients that target five sites on the surface protein, Env (Figure 4). The BNAbs with long CDR H3s typically target the V1V2-apex epitope that is generated in the quaternary structure of Env (Figure 4a). The V1V2 apex of Env is distal to the viral membrane and contains two relatively invariant features: The N-linked glycan sites at residues 160, 156 and 173, and a highly conserved net positive charge [29]. The basic amino acid side-chains in this region are thought to be critical for viral fitness as they facilitate conformational changes associated with Env binding to CD4 [50]. PG9 and P16 are archetypal V1V2 apex-targeting BNAbs, which potently neutralize cross-clade HIV-1 viruses in vitro via a wide, hammer-head-shaped CDR H3 that possesses a negatively charged specificity loop at its apex. This forms salt bridges with the positive loop at strand C of the V1V2 apex of Env and also contacts the surrounding glycans (Figure 4b,c) [15,28,29]. The most potent anti-HIV-1 BNAbs identified to date are those of the CAP256-VRC26 (CAP256) lineage, the members of which also target the positive region at the V1V2 apex of Env via a long CDR H3 loop derived from a VDJ rearrangement encoding a 35 amino acid CDR H3 [16]. Overall, 33 members of the CAP256 lineage have now been identified. However, in significant contrast to the PG9-like Abs, the neutralizing activities of the most effective CAP256 BNAbs are independent of the glycans surrounding the V1V2 site [16]. In these cases, the affinity maturation of the long CDR H3s has conferred a remarkable sequence independence, allowing the BNAbs to tolerate variability in their target. It appears that as long as the overall positive charge is maintained at the V1V2 apex, mutations elsewhere in the epitope are not detrimental to BNAb potency. Together, these Abs provide a proof of principle of the mechanisms by which extended CDR H3 structures can nullify evasive viral features by reaching into conformationally restricted regions of the viral glycoprotein [16,29]. These V1V2 directed BNAbs neutralize a large proportion of circulating HIV-1 strains and have the potential to be highly effective in protecting against HIV-1 transmission [51].

It is probable that similarly extensive cohorts of these ‘super-antibodies’ can be recovered from donors with previous exposure to other viruses [33]. For example, Ekiert et al. [30] used a phage-display library to isolate C05, a monoclonal BNAb capable of neutralizing diverse influenza A virus subtypes that protected against lethal viral challenge in mice [30]. The binding of C05 is mediated by a long 24 amino acid CDR H3 loop that inserts into the receptor binding site of the haemagglutinin glycoprotein trimer to neutralize viruses from many subtypes of influenza A. This loop-insertion mechanism generates exceedingly high-affinity contacts and is unlike any other previously documented antibody interaction. The single loop generated by the long CDR H3 targets a small, conserved epitope embedded in an otherwise hyper-mutable region [30,52].

More recently, the structure of a highly potent anti-Rabies virus BNAb bound to its epitope was described [53]. Here, the broad activity was generated by the binding of the BNAb to a highly conserved surface on the rabies virus glycoprotein G and locking G in its pre-fusion state. This means that the glycoprotein cannot undergo the conformational changes that are associated with the release of the viral genome into the cytoplasm upon exposure to the acidic pH of the endosome while the BNAb is bound. Though the CDR lengths were normal, the antibody was highly resistant to the mutagenesis of its epitope, with only mild losses in neutralization activity for single point mutations. While the heavy chain contributed 80% of the paratope surface, the SHM of both the light and heavy chains were found to be equally important for the breadth and potency of neutralization developed; in fact, the most pronounced reductions in activity were observed with amino acid changes at the light chain/heavy chain interface [53]. 

To date, the relationship between paratope and epitope has only been explored for one ultralong CDR H3. Ultralong CDR H3 variable exons were amplified from the lymphocytes of cows immunised with bovine viral diarrhoea virus (BVDV). These were then expressed and screened via ELISA and cell surface binding for affinity for BVDV. This strategy isolated the BLV1H12 clone. Its cysteine-folded knob structure is positively charged with multiple arginine residues and is entirely responsible for antigen binding. Mutational scanning revealed that the C-terminal residues of the knob are the most important for epitope binding, such that single arginine to alanine mutations here abolish the interaction, whereas increasing the positive charge in this region by removing acidic residues reduced specificity [18]. Further mutational analyses of the antigen binding sites of bovine ultralong CDR H3s will shed light on the unique kinds of interactions these novel receptors can potentially generate.

It is now clear that Abs with unusual paratope structures have an enormous potential for conferring broad protection, and the cow immune system may be a rich source of such candidate BNAbs. Surprisingly, BNAbs are rapidly induced following the immunization of cattle with a stabilized recombinant HIV-1 Env. Serum neutralisation breadth in the immunized cow was near total (96%) vs. a panel of 117 pseudo-typed, cross-clade viruses. Importantly, this included 63 clade A, B and C viruses that likely represent the most clinically relevant. This neutralizing activity could largely be recapitulated by a single isolated monoclonal Ab, Nc-cow1; this BNAb had a mean IC_50_ of 0.03 μg/mL and 72% coverage of the same 117 virus panel. Overall, this cow anti-HIV-1 antibody is amongst the most potent of the BNAbs discovered to date [20,51]. Remarkably, moderate BNAb activity in the serum was even observed after only six weeks [20]. This is considerably faster than the time taken to generate equivalent Abs during the natural course of HIV-1 infection in people. The use of a native-like recombinant Env trimer [54] likely contributed to the rapid induction observed in cows, since immunization with less native-like HIV-1 Env proteins only induced Abs with moderate potency. The use of the native-like immunogen in non-human primates (NHPs) has recently been shown to induce potent neutralizing Abs that provide protection against simian/human HIV-1 (SHIV) challenge [55] and the safety and utility of this immunogen is to be assessed in a Phase I clinical trial in humans (Trial Identifier: NCT03699241). However, competitive binding assays demonstrated that the monoclonal cow BNAbs targeted HIV-1 by engaging the CD4 binding site (CD4bs), which is a recessed, heavily occluded but highly conserved epitope that the equivalent human long CDR H3 Abs do not typically target. It is notable, however, that while BNAbs targeting the CD4bs are rare [51], this site was also vulnerable to neutralization by camelid ‘nanobodies’ via their long CDR H3 [21]. 

The most broad and potent of the camelid-derived heavy chain only nanobodies (VHH-A6) neutralised 80% of a diverse mixture of 15 clade A, B and C pseudotyped viruses with a mean potency of 2.6 μg/mL, but this could be significantly improved if two nanobodies were used in combination [21]. 

While, to the best of our knowledge, the specific neutralisation activity of the bovine ultralong CDR H3 has not been assessed against viruses other than HIV, it is of interest that polyclonal preparations of bovine immunoglobulins obtained from immunised cows can significantly reduce influenza A viral titres in the nose and lung of influenza-infected mice and can completely prevent infection if administered prophylactically. Future studies will help to elucidate the utility of this novel Ab type against other viruses with significant disease burdens [56].

Nonetheless, it is remarkable that these unusual antigen binding domains can target this heavily glycosylated receptor binding site of Env and exploit this distinct vulnerability in viral defences. These encouraging studies suggest that, together with camelid ‘nanobodies’, the bovine Ab repertoire might represent an untapped reservoir of potential BNAbs.

## 7. Summary and Future Directions

Broadly neutralizing antibodies with long CDR H3 are strong candidates for effective prophylaxis and antibody-based immunotherapy [33,51]. Bovine antibodies with their ultralong CDR H3s hold several notable advantages over other types of putative BNAb (Table 1) [19,20]. 

Nonetheless, growing evidence suggests BNAbs can provide a protective passive immunity that lasts for many weeks post-administration. Undoubtedly this could be improved by the rational mutation of the constant domains [34]. The rapid ex vivo engineering of the antibody variable domain is now also feasible. A recent study used a CRISPR-based mutagenesis strategy to generate a monoclonal Ab with nanomolar affinity from an antigen naïve pool within only three rounds of selection [57]. It is currently unknown if a similar strategy could generate effective antibodies from a library of bovine antibodies bearing ultralong CDR H3s. However, any ex vivo diversification strategies can primarily be focussed on the D segment, as most stabilizing structural features are already ‘hardcoded’ elsewhere in the heavy chain and light chain sequences. Indeed, these variable domains can be transplanted onto human Ab scaffolds whilst maintaining breadth and potency [20], which would circumvent any issues with host-generated anti-cow antibodies. Therefore, these ultralong CDR H3 may be very amenable to engineering for therapeutic purposes. Previous efforts to engineer the cow ultralong CDR H3 have already shown that the knob structure can be replaced by bioactive peptides to target specific receptors, including CXCR4 [58] or replaced with cytokines to generate highly functional fusion proteins [59]. The bovine antibody scaffold is thus also a versatile platform that has the potential to generate novel therapeutics with enhanced kinetics, including longer plasma half-lives. To our knowledge, no studies have yet asked whether bovine ultralong CDR H3 can be utilised in bi- or tri-specific antibodies. However, the possibility of being able to use the same light chain for multiple ultralong CDR H3s with different specificities may reduce issues with heavy chain/light chain pairing observed for some multi-specific antibodies. Studies of the Ab repertoires of malaria-infected individuals demonstrated that the entire extracellular domain of the LAIR1 receptor could be inserted at the CH1 hinge region. Remarkably, together with somatic mutations, this generated an endogenous broadly reactive Ab with two specific antigen binding domains on the same polypeptide chain [60]. Whether the ultralong cow ‘knob’ domains could be inserted at the CH1 region is yet to be explored.

Although anti-retroviral drug strategies have proven effective at lowering HIV-1 viraemia levels, this suppression is dependent on a strict dosing regimen and significant issues still persist with global drug availability. The targeted delivery of a prophylactic, BNAb-based intervention to vulnerable populations could be an effective method for preventing HIV-1 infections, especially as some BNAbs have been shown to confer protection that persists for months following treatment cessation [61]. A similar strategy can be exploited against other viruses, such as influenza or SARS-CoV-2 (Covid-19), removing the need for costly annual vaccines or providing a therapeutic response to pandemic outbreaks to stem community outbreaks. More speculatively, the application of a biological virucide containing these stable BNAbs directly at sites permissive to HIV-1 entry could provide a convenient method for limiting virus transmission. This is especially true since binding by ultralong CDR H3 BNAbs is maintained in acidic environments as low as pH 4 [20], as would be found on the vaginal mucosal membranes that represent one of the primary sites for HIV-1 entry.

Finally, it is clear that these novel structures can be utilised to bind to unique epitopes inaccessible to conventional antibodies. This understanding could expand antibody-based immunotherapies, as well as generate a plethora of new experimental tools. There is also a vast potential for these novel antibody types to target complex epitopes of therapeutic and biological interest.

## Figures and Tables

**Figure 1 viruses-12-00473-f001:**
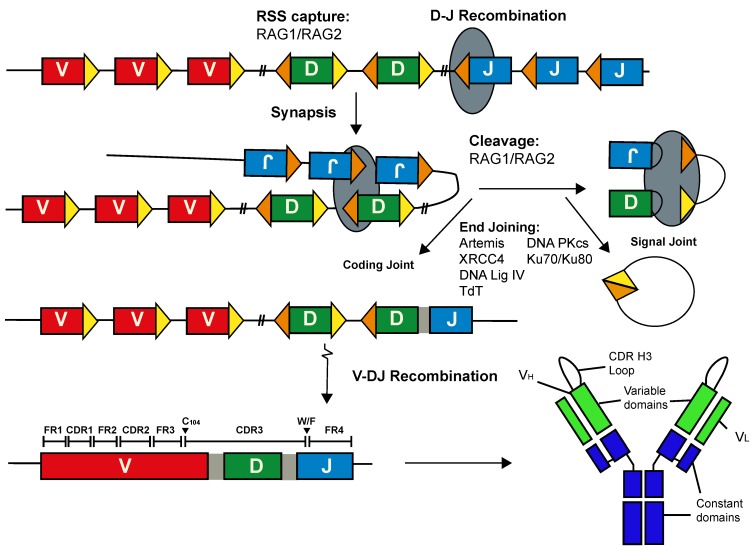
Assembly of Immunoglobulin Genes by V(D)J recombination. V(D)J recombination is a step-wise process that produces a contiguous variable exon from distinct germline segments. The complementarity determining region 3 of the heavy chain is encoded by the 3′ sequence of the V gene segment, the entire D gene segment and sequences at the 5′ end of the J segment. The framework and complementarity-determining regions are labelled, while the conserved features that demarcate the immunoglobulin heavy chain CDR H3 are indicated. The gaps between gene segments, shaded light grey, represent N-nucleotides. RAG1 and RAG2 proteins are depicted as a dark grey oval; proteins involved in end joining, Ku70/Ku80 heterodimer, Artemis, DNA protein kinase catalytic subunit (DNA-PKcs), Terminal deoxynucleotide transferase (TdT), X-ray repair cross complementing protein 4 (XRCC4) and DNA ligase IV, are indicated. A functional B-cell receptor is generated following light chain recombination and the subsequent pairing of the heavy (V_H_) and light (V_L_) chains. In some cases, V(D)J recombination can generate long CDR H3 loops that form unusual paratope structures.

**Figure 2 viruses-12-00473-f002:**
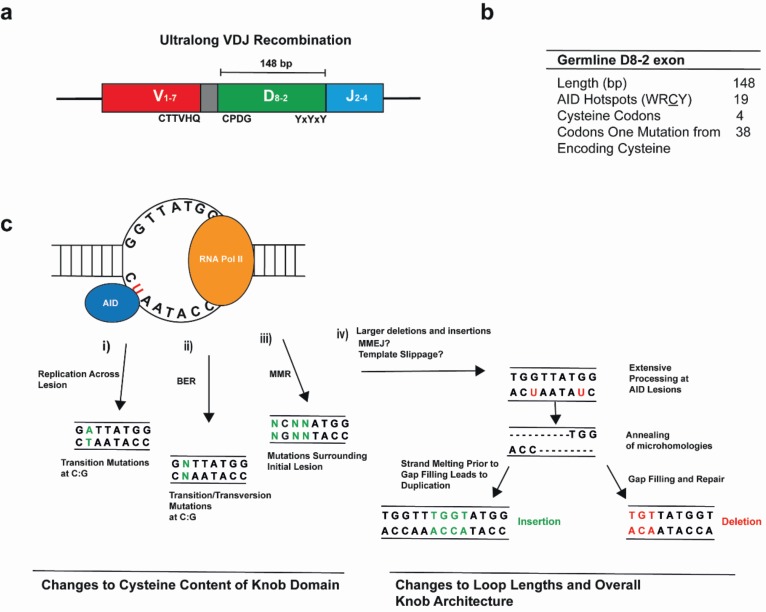
A single recombined variable exon is diversified to generate the ultralong repertoire in cows. (**a**) A schematic depicting the variable exon that encodes the ultralong CDR H3 of bovine antibodies. Conserved amino acids seen in the mature antibodies are indicated beneath the schematic. (**b**) The germline D8-2 gene segment encodes only four cysteines across its 148 bp sequence. However, the high density of putative activation-induced cytosine deaminase (AID) hotspots, and the large fraction of codons only a point mutation away from encoding cysteine, allow for the facile generation of cysteine codons during somatic hypermutation. (**c**) AID mechanisms of mutation. Green nucleotides indicate mutations; (i) deamination of cytosines to uracil can induce a mutation at C:G base pairs during replication; (ii) during base excision repair (BER), uracil DNA glycosylase can remove the uracil, creating an abasic site to induce mutation to any nucleotide following error-prone trans-lesion synthesis; (iii) base pairs surrounding the initial lesion can also be mutated in a process dependent upon mismatch repair (MMR); and (iv) large deletions and insertions occur in the bovine D8-2 exon. These may be the result of errors during replication across the AID-induced lesions due to template slippage or the error-prone end joining of AID-induced breaks (the latter outcomes are illustrated).

**Figure 3 viruses-12-00473-f003:**
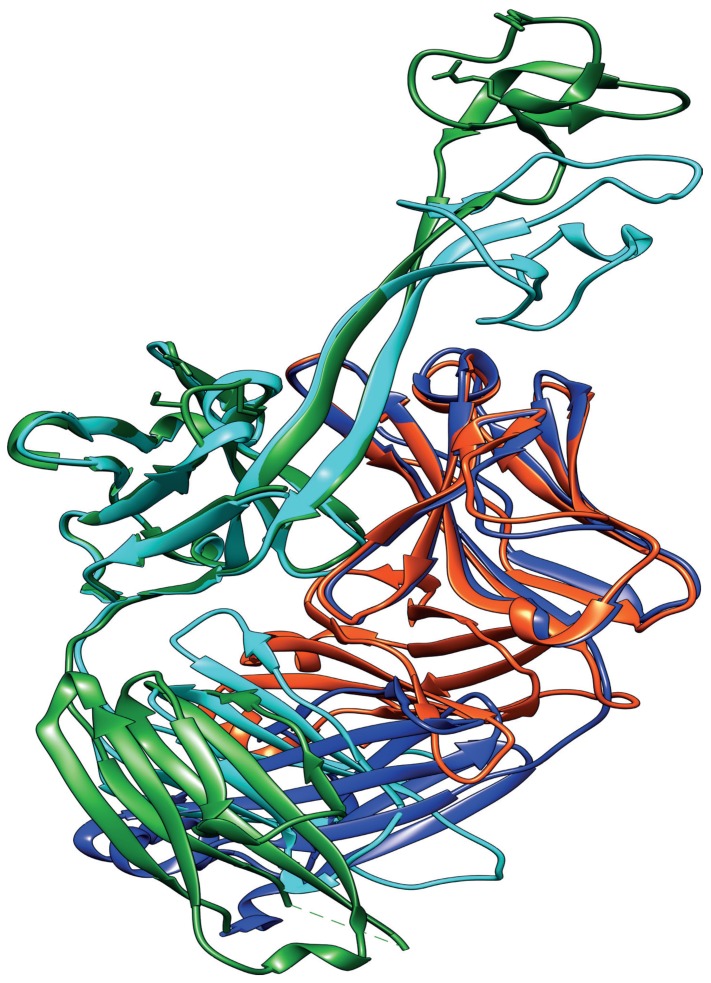
Diversity in the ultralong cow antibodies is focused in the knob domain. The overlay of the ribbon structures of two cow ultralong CDR H3 containing antibodies, BLV1H12 (heavy chain in green, light chain in dark blue; PDB: 4K3D) and Bov-7 (heavy chain in cyan, light chain in red; PDB: 6E9U). There is a 96% amino acid homology between the bulk of the two heavy chains, such that these regions of the overlaid structure have a root mean square error of 1.48 between equivalent α-carbons, as calculated by TopMatch [49]. In contrast, the knob domains are so divergent (<30% amino acid homology) that neither the amino acid sequences nor the 3D structures align successfully.

**Figure 4 viruses-12-00473-f004:**
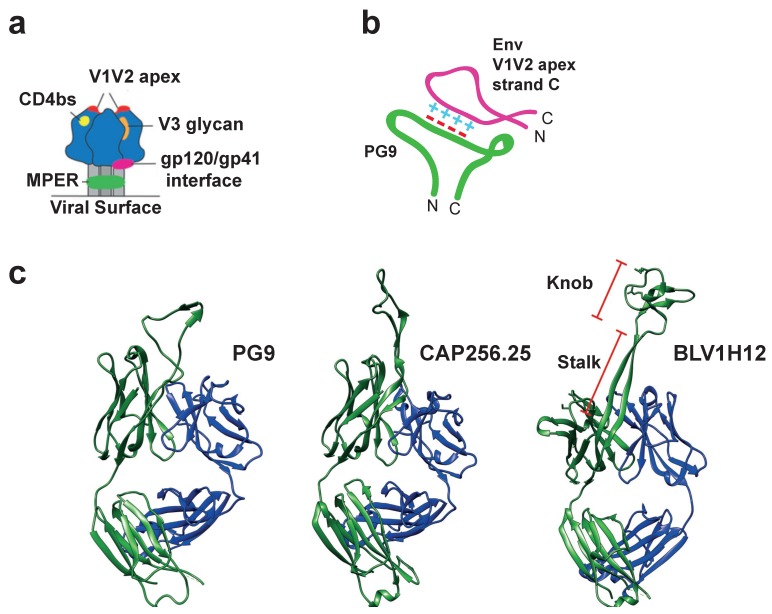
Extended CDR H3 structures generate unusual antigen binding sites. (**a**) A cartoon showing the five main epitopes on trimeric HIV-1 Env that are vulnerable to antibody-mediated neutralization. Depicted are: the CD4 binding site (CD4bs; yellow), membrane-proximal external region (MPER; green), the interface between gp120 and gp41 (pink), the V3 glycan supersite (orange) and the V1V2 apex (red), where the latter is the most distal from the viral surface. (**b**) A cartoon depicting the unusual structure of the CDR H3 of PG9 in contact with its epitope. The long anionic loop generates extensive electrostatic interactions with a conserved cationic region of HIV-1 Env. This interaction confers broad neutralization as long as the charge is conserved. (**c**) Structures of two human anti-HIV-1 BNAbs, PG9 (PBD: 5VJ6) and CAP256.25 (PDB: 5DT1) show that their CDR H3s extend away from the main antibody structure. The cow ultralong antibody BLV1H12 (PDB: 4K3D) is included for comparison, with the stalk and knob regions highlighted.

**Table 1 viruses-12-00473-t001:** Advantages and potential drawbacks of the ultralong CDR H3 structure.

Property	Advantage	Possible Drawbacks
Ultralong CDR H3 stalk and knob structure	Can reach conserved epitope targets to confer potent neutralization of broad viral variants (20). Knob domain can be replaced by bioactive peptides (59; 58)	Novel antigen binding site may generate anti-idiotype antibodies when used therapeutically in man.
Knob domain is highly stable	Able to maintain specificity and affinity after transplant onto scaffolds (20). Strong binding even in typically unfavourable conditions e.g., acidic pH (20).	Unknown if neutralization is also maintained at acidic pH.
IGHD8-2 gene encoding the knob domain is primed for diversification	Expansive antibody repertoire from only a single VDJ rearrangement (18).	It has not been shown how rapidly the IGHD8-2 gene can be mutated ex vivo to generate insertions, deletions, and cysteine codons.
All other CDRs are invariable	Affinity maturation process focussed solely on the CDR H3 (19).	Repertoire diversity relies on extensive AID activity prior to antigen exposure. If the VDJ is knocked-in to a different mammalian system, diversity in the antigen naïve repertoire may not be sufficient.

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
