# Peer review of "Broadly Neutralizing Bovine Antibodies: Highly Effective New Tools against Evasive Pathogens?"

_viruses, 2020, doi:10.3390/v12040473_

Round 1
Reviewer 1 Report
In this manuscript the authors review on the characteristics of bovine antibodies that distinguish them from classical immunoglobulins. Although some viruses targeted by such bovine antibodies are also addressed here, I think that this review is better suited for the MDPI sister journal MDPI Vaccines to address more the vaccine readership.
Generell comments:
Generally the review is well written and nicely summarizes the prominent special characteristics of bovine antibodies in the context of broad neutralization capacity. However, I missed a comparison with camelid nanobodies, which correspond to the variable region of single chain only antibodies in the camelid family, and have very similar features, in particular the capacity to also build long HCDR3 regions. These nanobodies have also been shown to potently neutralize a broad spectrum of HIV-1 subtypes by several groups by similar mechanisms (penetration into receptor cavities or the glycan shield). This should be mentioned. There is also a review by Muyldermans & Smider (2016) on bovine and camelid antibodies, which should be cited.
Chapter 1 of the introduction can probably be shortened, as it mostly contains basic immunological knowledge on antibody development and affinity maturation.
Minor comments:
- 1, line 12/13: “continued pharmacological intervention” -> I would primarily see the advantage of derivatives of such special antibodies in prevention rather than in therapeutic treatments, which is also meant in the second part of the sentence concerning Influenza vaccination.
P 3, Fig. 1 last row: in the immunoglobuline the variable regions should be indicated too like the constant regions. In addition there is a mistake, as the VL has to be moved up to the green part of the light chain.
If Artemis and Ku70/Ku80 are mentioned in Fig. 1 they should also be mentioned in the legend/text( or being skipped from the fig.).
p.4, line 87: …generate a vast Ab repertoire. Some viruses, including……
p.4, line 90: …or occlude epitopes that are…
p.4, line 99: here or after the next sentence, the camelid antibodies should be mentioned too.
p.4, line 112: ….of reaching hidden epitopes…
p.4, line 114/115:….C104 and W/F should be indicated in Fig. 1
p.5, line 132: …It is notable that many individuals….(there are other mechanisms too)
- 5, line 146: ….is not conductive to their induction via classical vaccination,….
p.6, line 190: DSB has not been explained before
p.7, line 201: difference instead of departure
p.11, line 318: …in significant contrast to…
p.12, line 370: …simian/human HIV-1 (SHIV)….
P12., line 375: mention that camelid nanobodies with long HCDR3 have been described that are able to target the CD4 binding site.
p.14, line 411-413: mention problem of potential anti-cow antibodies or state why this is not a problem.
p.14, line 422: …virucide…
p.14, line 424-426: neutralization at acidic pH has not been shown in the cited reference, just binding. Neutralization has been show for nanobodies at acidic pH though.
References: the paper on cow antibodies from by Vadnais and Smider 2016 should be cited too
Author Response
General comments:
We believe that this manuscript is better suited to Viruses than Vaccines because this is a very important area for Virologists who are studying conserved epitopes. Furthermore, it is important that this wider community is aware of these broadly neutralising antibodies, the mechanism that leads to their formation and the nature of their interactions with viral particles/epitopes.
We have now included references to camelid nanobodies (lines 96-98 & 379-381) and mentioned their similarities to bovine broadly neutralising antibodies. We have also cited the corresponding review (lines 98 & 381).
We have shortened the introduction slightly (lines 31 to 106) but we believe it is important to keep a reasonably broad introduction so that the review is accessible to as many virologists as possible.
Minor comments:
We have made the typographical changes suggested by the Reviewer on lines 87, 90, 112, 132, 146, 190, 201, 318, 370 and 422. These lines are now slightly shifted due to the other changes.
We have modified the text on lines 12/13 to add the use of broadly neutralising antibodies in prevention as well as therapeutic treatments.
We have modified Figure 1 as requested by:
- Labelling the variable region as we had done for the constant regions
- Correcting the labelling error for VL (and thank the Reviewer for pointing this out)
- Mentioning Artemis, Ku70/Ku80 and DNA ligase IV in the Figure legend
- Indicating C104 and W/F as requested (Reviewer’s comment, line 114/115, now 112/113).
As suggested, we have pointed out that camelid nanobodies with long HCDR3 can target the CD4 binding site (lines 379-381) and thank the Reviewer for suggesting that we include this.
Line 424-426: we have mentioned the potential of anti-cow antibodies being generated but explained that this is not a problem as the ultralong variable domains can be grafted onto human constant domains.
Line 452-453: we have clarified that it is binding by ultra-long CDR H3 broadly neutralising antibodies that has been shown to occur at acidic pH.
The paper by Vadnais and Smider was actually already referenced (Reference 44, now 48).
Reviewer 2 Report
This review titled “Broadly neutralizing bovine antibodies: highly effective new tools against evasive pathogens” focuses on the structural aspects of cow antibodies and potential advantages of the long bovine CDRH3 region for recognition of pathogen antigens, specifically viral surface glycoproteins. It is well-written and thought-provoking, but I do have some comments for the authors to contemplate and address:
- In Figure 2, how a deletion might occur is illustrated, but that would not lead to a longer CDRH3 region. It would be helpful to illustrate how an insertion could occur, with possible mechanisms, since that is likely of greater import.
- It is not clear from the narrative whether the differences in length of the CDRH3 is due to a limitation in man (such that the region is shorter) vs. an absence of limitation in cows (such the region is longer). This could be addressed in part by comparing lengths in various mammalia.
- The authors discuss the structural details of the CDRH3 of bovine Abs, which is helpful. It would also be very useful if the authors provided functional data regarding these Abs, such as IC50or IC90values (neutralization potency), binding affinities and kinetics if known, how broad they are (especially with regards to HIV strains/isolates), and what other pathogens (other than HIV) they have been used to target (I believe the rabies one mentioned is human. One obvious question would be whether these Abs have any utility for any other pathogen, other than HIV?
- The authors discuss some engineering of cow Abs—but can they be made into bi- or tri-specific Abs? Has it ever been tried? Or, for example, grafting the long CDRH3 region onto a human AB?
- The authors describe in detail the long CDRH3 structure that targets the HIV Env V1V2-apex epitope. What about other Env epitopes?Do those get recognized by cows?
- I would suggest expanding the table (and including in the text) listing and comparing additional advantages vs. disadvantages of cow vs. human Abs. An obvious disadvantage not discussed is that the cow ones should generate anti-isotype and anti-idiotype antibodies, which will greatly limit their utility in man.
Author Response
- We have re-drawn Figure 2 to show the possible mechanisms by which insertions could occur, as requested.
- We have compared CDR H3 lengths in different mammals as requested. From the available sequences, the ultra-long bovine CDR H3 appears to be due to the unique genetics of bovine and, to an extent of the Camelidea. Indeed, most other mammalian CDR H3 lengths appear to be similar, or slightly shorter than human. We have added a sentence to this effect (lines 168-172).
- We have now added some sentences (lines 362-368) to provide functional data for the bovine broadly neutralising anti-HIV antibodies by giving both the potency data and the breadth of their neutralisation. We have also compared the potency of the bovine antibodies with the camelid-derived nanobodies (lines 382-385). Although the specific neutralisation activity of the bovine ultralong CDR H3 has not been assessed against viruses other than HIV, we describe initial data pertaining to the efficacy of bovine anti-influenza A antibodies (lines 386-392), which indicates that these antibodies have the potential to be used against other viruses.
- As far as we know, no studies have attempted to make bi- or tri-specific involving bovine ultralong CDR H3. We have however, now discussed this intriguing possibility in the future directions (lines 432-441) and thank the Reviewer for this suggestion.
- Bovine anti-HIV antibodies have been shown by competition ELISA to bind to the CD4 binding site of HIV but a detailed cryoEM structure of this binding has not been generated. Likewise, bovine antibody binding to other Env epitopes has not been explored.
- In response to Reviewer 1, we have already discussed the possibility that antibodies against cow isotypes can be overcome by grafting the ultralong variable domains onto human constant domains (lines 424-426). In addition, we have now outlined the possible drawbacks of the ultra-long CDR H3 structure, including anti-idiotype antibodies, in the Table (lines 459-460) and thank the Reviewer for suggesting this.
Reviewer 3 Report
This is a well written review on bovine antibodies. I enjoyed reading it.
Author Response
We thank the Reviewer for their very positive review.
Reviewer 4 Report
Well written and structured review.
Author Response

(The authors gave the same response as above.)
